# Sex-Related Differences in Cerebrospinal Fluid Plasma-Derived Proteins of Neurological Patients

**DOI:** 10.3390/diagnostics11050884

**Published:** 2021-05-16

**Authors:** Massimiliano Castellazzi, Caterina Ferri, Sarah Alfiero, Ilenia Lombardo, Michele Laudisi, Ginevra Tecilla, Michela Boni, Stefano Pizzicotti, Enrico Fainardi, Tiziana Bellini, Maura Pugliatti

**Affiliations:** 1Department of Neuroscience and Rehabilitation, University of Ferrara, 44121 Ferrara, Italy; caterina.ferri@unife.it (C.F.); sarahalfiero.98@gmail.com (S.A.); ilelombardo1@gmail.com (I.L.); ldsmhl@unife.it (M.L.); tclgvr@unife.it (G.T.); tiziana.bellini@unife.it (T.B.); maura.pugliatti@unife.it (M.P.); 2Interdepartmental Research Center for the Study of Multiple Sclerosis and Inflammatory and Degenerative Diseases of the Nervous System, University of Ferrara, 44121 Ferrara, Italy; 3Chemical-Clinical Analysis Laboratory, “S. Anna” University Hospital, 44124 Ferrara, Italy; m.boni@ospfe.it (M.B.); s.pizzicotti@ospfe.it (S.P.); 4Department of Experimental and Clinical Biomedical Sciences, University of Florence, 50121 Florence, Italy; henryfai@tin.it; 5University Center for Studies on Gender Medicine, University of Ferrara, 44121 Ferrara, Italy

**Keywords:** sex, age, cerebrospinal fluid, albumin quotient, intrathecal IgG synthesis, blood-cerebrospinal fluid barrier, humans

## Abstract

Background and aims: Cerebrospinal fluid (CSF) protein content presents a sexual dimorphism in humans. We investigated sex-related differences in CSF IgG levels and in the quantification of intrathecal IgG synthesis (IIS). Methods: CSF, serum albumin and IgG were measured in 1519 neurological patients and both linear and hyperbolic formulas were used for the quantification of IIS. CSF-restricted oligoclonal IgG bands (OCBs) were used as “gold standard”. Results: The linear IgG Index showed a weak agreement with OCBs in males and females (k = 0.559, k = 0.587, respectively), while the hyperbolic Reiber’s formulas had a moderate agreement with OCBs in females (k = 0.635) and a weak agreement in males (k = 0.565). Higher CSF albumin and IgG levels were found in men than in women in the whole population and in subjects without IIS after adjusting for age and for serum concentrations of albumin and IgG, respectively (Quade statistics, *p* < 0.000001). CSF and serum albumin and IgG levels positively correlated to age in both sexes. CSF total protein content did not correlate with CSF leukocyte numbers but was higher in patients with marked pleocytosis. Conclusions: In neurological patients, men have higher levels of CSF serum-derived proteins, such as albumin and IgG.

## 1. Introduction

Cerebrospinal fluid (CSF) is a watery body fluid that is continuously produced by the choroid plexuses of the cerebral ventricles and circulates around the surface of the brain and spinal cord. CSF typically contains low protein concentrations and few circulating cells [1]. In addition to its hydromechanical protective function, CSF plays a leading role in regulating the homeostasis of the central nervous system (CNS) [2].

CSF studies open “windows on the brain” [3]. The diagnostic lumbar puncture, or spinal tap, is a mini-invasive procedure used to withdraw CSF from the spinal subarachnoid space. The CSF analysis represents a fundamental tool to investigate the presence of infectious, inflammatory and degenerative conditions within the CNS [4]. 

The most abundant proteins in blood serum are albumin and immunoglobulin G (IgG), approximately 40 and 10 mg/mL, respectively [5]. Albumin is used as a reference protein for the blood–CSF barrier (B-CSF-B) as it originates mainly from the blood [6]. Serum and CSF albumin determinations, including the CSF/serum albumin quotient (named “QAlb”), allow the degree of B-CSF-B permeability (the “B-CSF-B damage” at the level of the choroid plexus, different from the blood–brain barrier damage at the level of blood vessels that vascularize the CNS) to be measured [7,8]. The combination of these parameters with CSF and serum concentrations of IgG, through mathematical formulas, enables the determination of whether CSF IgGs are being at least partially produced intrathecally or are all plasma-derived [9]. In the clinical-laboratory practice, the most used mathematical formulas are the linear IgG Index [10] and the hyperbolic Reiber’s formula [11].

Sexual dimorphisms have been reported related to brain development, structure and neurotransmission [12], and which can predispose to sex-specific neurological conditions [13]. Recent studies on QAlb, have shown sex differences in B-CSF-B permeability in both hospital and general populations [14], in multiple sclerosis, and other inflammatory and noninflammatory neurological disorders [15] and in patients with schizophreniform and affective psychosis [16]. Such differences in the QAlb concentration produces a sexual dimorphism of the CSF protein concentrations wherein males have higher protein levels than females, independently of age [17,18].

While it is clear that the concentration of plasma-derived proteins, such as albumin, is higher in the CSF of males than in females, the sex-specific concentration of immunoglobulins, another main component of plasma proteins, still remains unknown.

We therefore aimed to search for possible sex differences in the CSF content of plasma proteins, with a focus on albumin and IgG, and to analyze the impact that these differences may have on the application of the most widely used mathematical formulas for measuring B-CSF-B damage and the intrathecal synthesis of immunoglobulins.

## 2. Materials and Methods

### 2.1. Study Design

Clinical and laboratory anonymized data were retrospectively collected from patients hospitalized from 2000 to 2018 at the “S. Anna” University Hospital (Azienda Ospedaliero-Universitaria S. Anna), in Ferrara, northern Italy. The study was approved by the local Committee for Medical Ethics in Research, “Comitato Etico di Area Vasta Emilia Centro della Regione Emilia-Romagna” (Prot. N. 770/2018/Oss/AOUFe, dated 12/12/2018) and written informed consent was obtained.

### 2.2. Cohorts

Two different cohorts of patients were included in this study. 

Cohort 1 were used for the analysis of CSF and serum albumin and IgG levels and for the determination of an intrathecal IgG synthesis. This group included neurological patients of which almost a quarter were multiple sclerosis-affected people, a quarter of the patients with other inflammatory neurological diseases, a quarter of the patients with non-inflammatory neurological diseases and the last quarter were people with undefined diagnosis as published before [15]. 

Cohort 2 were used as the control group to study the relationship between CSF total protein content and pleocytosis and included patients who underwent lumbar puncture for diagnostic purposes which had (i) normal (up to 5 cells/μL), (ii) mild (6–25 cells/μL), (iii) moderate (26–100 cells/μL) and (iv) marked pleocytosis (more than 100 cells/μL).

All patients in both cohorts were analyzed blindly with respect to diagnostic suspicion and definite clinical diagnosis.

### 2.3. Pre-Analytical Procedures

Pairs of CSF and blood samples were withdrawn from all patients included in the study for diagnostic purposes. CSF and serum samples were analyzed at room temperature immediately after centrifugation or stored in aliquots at −80 °C until assay. Exclusion criteria were: age less than 16 years, lack of demographic information (sex and/or age) and the presence of xanthochromia and/or high levels red blood cells (>2000/µL) in CSF, as suggestive for traumatic lumbar puncture or subarachnoid hemorrhage.

### 2.4. Samples Analysis

#### 2.4.1. Cohort 1

All analyses were performed on the 2nd–4th mL of CSF after lumbar puncture [15].

Albumin and IgG levels were measured as part of the diagnostic work-up in cell-free CSF and paired serum samples by immunochemical nephelometry with the Beckman Array Protein System or IMMAGE 800 Immunochemistry System (Beckman Instruments, Fullerton, CA, USA) following the procedure of Salden [19]. Albumin quotient was calculated to disclose B-CSF-B dysfunction according to the formula: QAlb = [albumin]_CSF_/[albumin]_serum_ × 1000

Normal QAlb values were considered as <6.5 for patients aged 15–40 years, <8.0 for patients aged 41–60 years and <9.0 for patients over 60 years [9,20,21]. Accordingly, QAlb was considered abnormal for values greater than or equal to the reported thresholds.

Quantification of the IIS was done with both linear and hyperbolic mathematical formulas as follow:(1) linear IgG Index = QIgG/QAlbwhere QIgG = [IgG]_CSF_/[IgG]_serum_ × 1000(2) the hyperbolic Reiber’s formula, IgGLoc = (QIgG − QLim) × [IgG]serumwhere QLim $=0.93\times \sqrt{QAlb^{2}+6\times 10^{-6}-1.7\times 10^{-3}}$

The normal limits were considered 0.7 and 0 for IgG Index and for Reiber’s formula, respectively.

Qualitative demonstration of the IIS was done on paired CSF and serum samples with the “gold standard” of isoelectric focusing on agarose gel followed by IgG-specific immunoblotting using a Helena Biosciences IgG IEF kit (Helena Biosciences Europe, Gateshead, UK). The immunoblotting patterns were first independently assessed by two experienced operators blinded each other and for clinical information, and finally the presence/absence of IgG oligoclonal bands (OCBs) were established through an interobserver agreement. The following CSF patterns were considered: pattern 1 = absence of IgG OCBs; pattern 2 = 2 or more CSF-restricted IgG OCBs; pattern 3 = 2 or more CSF-restricted IgG OCBs with additional identical IgG OCBs in CSF and serum; pattern 4 = 2 or more identical IgG OCBs in CSF and serum; pattern 5 = some identical IgG OCBs in CSF and serum in a restricted pH range. Only patterns 2 and 3 indicated an intrathecal IgG synthesis [9].

#### 2.4.2. Cohort 2

CSF total protein content was measured in all samples as a part of the diagnostic work-up. As published before [18] analysis was performed on the 1st–4th mL of CSF after lumbar puncture within 2 h from sample withdrawal. CSF samples were analyzed at room temperature, immediately after centrifugation and with the Beckman Coulter AU640/AU640e automated chemistry analyzer (Beckman Instruments, Fullerton, CA, USA). The lower detectable analyte level was estimated to be 7 mg/dL, with a sample interval of 1–199 mg/dL.

CSF leukocytes were counted in a Fuchs-Rosenthal chamber and counts were reported as cells to a standard volume of 1 μL. A cell count less than or equal to 5 cells/μL was considered normal, higher values were considered suggestive of pleocytosis.

### 2.5. Statistical Analysis

Continuous variables presenting with a non-normal distribution (Kolmogorov–Smirnov test) were reported as median and interquartile range (IQR) and Mann–Whitney test was used for all comparisons. Categorical variables were reported as counts (percentages) and the Fisher’s exact test was used to test significance. Correlations between age and biological variables were investigated with the Spearman test. In regression analysis F-test was used to compare the fits of linear models. The agreement with quantitative and qualitative methods for the determination of an IIS was calculated through the Cohen’s kappa coefficient [22] with a free available tool:

https://www.graphpad.com/quickcalcs/kappa1/. 

Kappa values were interpreted as follows: 0.00–0.20, slight agreement; 0.21–0.40, fair agreement; 0.41–0.60, moderate agreement; 0.61–0.80, substantial agreement; 0.81–1.00, almost perfect agreement. The sensitivity and specificity of the mathematical formulas in identifying the presence of IIS were calculated with respect to the “gold standard” of OCBs using 2 × 2 contingency tables. The robust regression and outlier removal (ROUT) method was used to determine outliers, assuming a maximum desired false discovery rate (Q) equal to 1%. Quade statistics was used for multivariate analysis. Two tailed *p*-values < 0.05 were considered statistically significant. The Statistical Package for the Social Sciences (IBM SPSS Statistics for Windows, Armonk, NY, USA) version 21.0 for Windows and Prism 9 for MacOS (GraphPad Software, La Jolla, CA, USA) were used for the statistical analysis. 

## 3. Results

### 3.1. Patients’ Characteristics

Cohort 1 included 1519 patients (859 females and 660 males) aged between 15 and 93 (Table 1). The women:men ratio was 1.30 in the overall population. Median age at admission was higher in men (50 years) than in women (43 years). CSF median levels of IgG and albumin were more elevated in men than in women (4.63 vs. 3.59 mg/dL and 29.7 vs. 19.2 mg/dL, respectively) with no differences between sexes for IgG and albumin in serum samples. Albumin quotient median value was higher in men than in women (7.50 vs. 4.93) while an intrathecal IgG synthesis was more frequently detected in women than in men both with the quantitative methods (positive IgG Index and Reiber’s formula: 36.0 vs. 24.4% and 36.7 vs. 19.2% respectively) and the qualitative determination of OCBs (40.4 vs. 23.7%).

### 3.2. Agreement Between Quantitative Method and the “Gold Standard” of Oligoclonal Bands for the Determination of an Intrathecal IgG Synthesis

The linear “IgG Index” showed a moderate agreement both in females (k = 0.587) and males (k = 0.559) while the hyperbolic Reiber’s formula demonstrated a good agreement in females (k = 0.635) and a moderate agreement in males (k = 0.565) (Table 2).

Sensitivity and specificity of IgG index and hyperbolic Reiber’s formula were calculated and compared in the two sexes (Table 2). While the linear IgG Index showed similar performances in both females and males with no statistical differences for sensitivity and specificity, the hyperbolic Reiber’s formula had higher sensitivity in women than in men (73 vs. 60%) with, on the other hand, better specificity in men than in women (93 vs. 90%).

### 3.3. Correlations between Age, Sex and CSF and Serum Concentrations of Albumin and IgG

CSF and serum concentrations of albumin and IgG and their CSF/serum quotients were investigated by age and sex. Patients with IgG intrathecal synthesis (OCBs patterns 2 and 3) were excluded from this analysis to avoid the influence of the intrathecal IgG fraction on the blood-derived CSF IgG concentration and consequently on the CSF/serum IgG quotient. Moreover, to reduce potential biases related to abnormal CSF or serum concentrations, outliers were excluded from any single analysis.

In linear regression analysis, CSF albumin concentrations were differently associated to age in females and males (difference between slopes: F = 4.720, *p* = 0.031; the difference between Y-intercepts was undeterminable for too large difference of slopes) with a positive correlation (Spearman: r = 0.2218 and *p* < 0.0001) in females and with no correlation in males (Spearman: r = −0.0004, *p* = 0.9926) (Figure 1, panel a). CSF IgG levels showed a similar association with age in both females and males with higher levels in males at any age (difference between slopes: F = 1.890, *p* = 0.1696; difference between Y-intercepts: F = 65.85, *p* < 0.0001) and a positive correlation in females (Spearman: r = 0.2145, *p* < 0.0001) and in males (Spearman: r = 0.09924, *p* = 0.0368) (Figure 1, panel b). Serum albumin concentrations fell more rapidly with age (difference between slopes: F = 13.20 and *p* = 0.0003; difference between Y-intercepts was undeterminable for too large difference of slopes) in males (Spearman: r = −0.4778, *p* < 0.0001) than in females (Spearman: r = −0.2614 and *p* < 0.0001) (Figure 1, panel c) while serum IgG levels showed a similar relationship to age in the two sexes (differences between slopes: F = 0.9380, *p* = 0.3330; difference between Y-intercepts: F = 0.1100, *p* = 0.7403) despite a slightly higher negative correlation in females (Spearman: r = −0.1307, *p* = 0.0032) than in males (Spearman: r = −0.03953, *p* = 0.3782) (Figure 1, panel d). QAlb was positively correlated to age both in females (Spearman: r = 0.3033, *p* < 0,0001) and males (Spearman: r = 0.1525, *p* = 0.001) and the regression line was constantly higher in males than in females (differences between slopes: F = 0.3589, *p* = 0.5493; difference between Y-intercepts: F = 135.1, *p* < 0.0001) (Figure 1, panel e). Likewise, QIgG positively correlated with age, both in females (Spearman: r = 0.2813, *p* < 0.0001) and males (Spearman: r = 0.1460, *p* = 0.002) with a higher regression line in the male group (difference between slopes: F = 0.2055, *p* = 0.6504; difference between Y-intercepts: F = 106.5, *p* < 0.0001) (Figure 1, panel f).

### 3.4. Age-Matched Comparison of Cerebrospinal Fluid Albumin and IgG Levels in Patients with no Evidence of Intrathecal IgG Synthesis

As for the previous analysis, patients positive for an IgG intrathecal synthesis (OCBs patterns 2 and 3) were excluded to avoid the influence of the intrathecal IgG fraction. CSF albumin and IgG levels were compared in men and women grouped by age. As reported in Figure 2, male patients had higher CSF levels of both albumin and IgG in all age ranges considered.

### 3.5. Multivariate Analysis

A multivariate model was applied to patients negative for the presence of an intrathecal IgG synthesis. As reported in Table 3, CSF albumin (F = 83.329, *p* < 0.000001) and IgG (F = 52.827, *p* < 0.000001) median levels were significantly higher in men than in women even after adjusting for age at the time of lumbar puncture and for serum levels of both albumin and IgG.

### 3.6. Association between Cerebrospinal Fluid Cellular and Protein Content

Cohort 2 included 577 subjects, 299 women and 278 men, grouped by pleocytosis severity in four groups: (i) normal (no more than 5 cells/μL), (ii) mild (6–25 cells/μL), (iii) moderate (26–100 cells/μL) and (iv) marked (more than 100 cells/μL). All groups had a similar M/F ratio with no differences in age between sexes (data not shown). As reported in Figure 3, men with a marked pleocytosis had higher CSF protein levels than patients with a normal pleocytosis (Figure 3, panel a), while women with marked pleocytosis had higher protein levels in CSF than women with normal, mild and moderate pleocytosis. 

CSF protein levels were further correlated to the number of CSF infiltrating cells in both men and women with mild, moderate and marked pleocytosis but no associations were found (Table 4).

## 4. Discussion

In this study, we highlight for the first time that CSF concentrations of serum-derived IgG, similar to that already described for albumin, are higher in male than in female neurological patients and that these concentrations increase with age in a parallel manner in the two sexes. These differences in CSF composition have an impact on the applicability of the quantitative indices of IIS as shown with the differences between sensitivities, specificities and agreement with the “gold standard” of OCBs by the Reiber’s hyperbolic formula.

The CSF analysis is a widely used laboratory tool supporting the diagnosis of neurological disorders. In particular, quantitative CSF analysis, through the measurement of albumin and IgG in paired serum and CSF, together with the subsequent calculation of mathematical indices, can give information regarding the B-CSF-B functionality and the presence of inflammation within the CNS [23,24]. The measurement of albumin and IgG by nephelometry is a fast, reproducible, automated, and therefore operator-independent tool for the biochemical evaluation of CSF. It is generally thought that all albumin measured in CSF is of plasma derivation and for this reason the QAlb was considered a valid support for the evaluation of B-CSF-B functionality [25]. The albuminocytologic dissociation, characterized by an elevation of CSF total protein as well as by an increase of QAlb, is a characteristic of the acute phase of Guillain Barrè syndrome [26]. Moreover, increased QAlb was associated with higher brain atrophy and greater disability in clinical onset MS [27] and is considered a potential indicator of disease severity in newly diagnosed neuromyelitis optical spectrum disorder [28]. For decades QAlb was considered directly associated with age, and it has been recently shown that sex can also modulate it. Higher QAlb were reported in males than in females in different neurological patient cohorts: in MS patients and in inflammatory and noninflammatory neurological diseases [15], in patients with psychiatric disorders [16], and in a large group of not better characterized neurological patients, as well as in some subgroups of healthy controls [14].

In the present study we also demonstrated that the different CSF concentrations of plasma-derived proteins not only concerned albumin, but also IgG, the second most abundant plasma protein. A consequence of this sex-specific distribution of albumin and IgG was detected in the different performance of the quantitative indices of IIS in females compared to males. In particular, the hyperbolic Reiber’s formula showed sex-related differences in sensitivity, specificity and concordance towards the “gold standard” of OCBs. To exclude the effect of the intrathecally produced IgG fraction on the CSF, total IgG concentration in all patients with an IIS by means of IgG OCBs were excluded from subsequent analyses. Surprisingly, serum IgG levels remained constant and were similar with age in the two sexes, while CSF IgG levels were higher in males and increased with age in a similar way in both sexes. The consequence of this sexual dimorphism in CSF protein content was that the CSF/serum IgG quotient (QIgG) increased in parallel in the two sexes, maintaining higher levels in males than in females at any age, similar to what has been reported about QAlb. Taken together, this evidence highlights a higher concentration of plasma-derived proteins, i.e., albumin and IgG, in males due to a greater permeability of B-CSF-B, as shown by the differences between the respective CSF/serum quotients.

The higher CSF concentration of plasma-derived proteins in males compared to females is reflected in that males tend to have a higher CSF total protein content than females [18]. Different positivity thresholds have already been proposed for males and females with regards to the CSF total protein content [17] and their application effectively reduces the gap between the two sexes in laboratory reporting of high protein content [18,29].

Interestingly, in both sexes a marked pleocytosis is accompanied by a high CSF protein content while this association is not present for less severe pleocytosis. Furthermore, the CNS infiltrating lymphocyte count does not appear to correlate directly with CSF protein concentration values in our neurological patients.

Due to the observational nature of our study, we are unable to draw conclusions on the causes of this different CSF biochemical composition. Sex-related differences in CSF composition may be secondary to multiple mechanisms. The female hormone 17β-estradiol [30] could drive a different expression of the enzymes involved in the turnover and breakdown of the barrier [31,32], essentially having a protective role with respect to breaking of the barrier [33]. Consequently, since the sex-specific values of QAlb and total protein content of CSF do not change during puberty or menopause, the role of genetic predisposition linked to sex chromosomes, together with the role of the hormones themselves, should be considered [14,15]. Moreover, differences in weight, height and body mass index between males and females could also influence the CSF protein composition [34].

In light of these results, it is evident that the study design of any CSF biomarker, especially if of possible (partial or total) plasma derivation, must necessarily take into account the effects of sex and age.

Our study has some limitations. Our analyses did not include a population of controls from the general populations, in relation to the lumbar puncture practice only performed for diagnostic or therapeutic purposes. Moreover, the lack of clinical diagnoses did not allow us to verify the impact of this sex-related discrepancy in those conditions such as MS in which the presence of IIS is one of the diagnostic criteria. Finally, due to the retrospective nature of the study, it was not possible to include some physical variables such as weight, height and body mass index in our analyses.

In conclusion, our study confirms the presence of a sexual dimorphism with regard to CSF protein content and suggests that this is mainly due to higher levels of plasma-derived proteins in male CSF. The effects of this gap between the two sexes fall on the quantitative indices of IIS and evaluation of the functionality of B-CSF-B. Future studies are desirable, and especially on populations of subjects with no neurological disorders, to provide indications on possible corrections to apply to the quantitative parameters of CSF analysis in order to standardize the physiological gap between the two sexes.

## Figures and Tables

**Figure 1 diagnostics-11-00884-f001:**
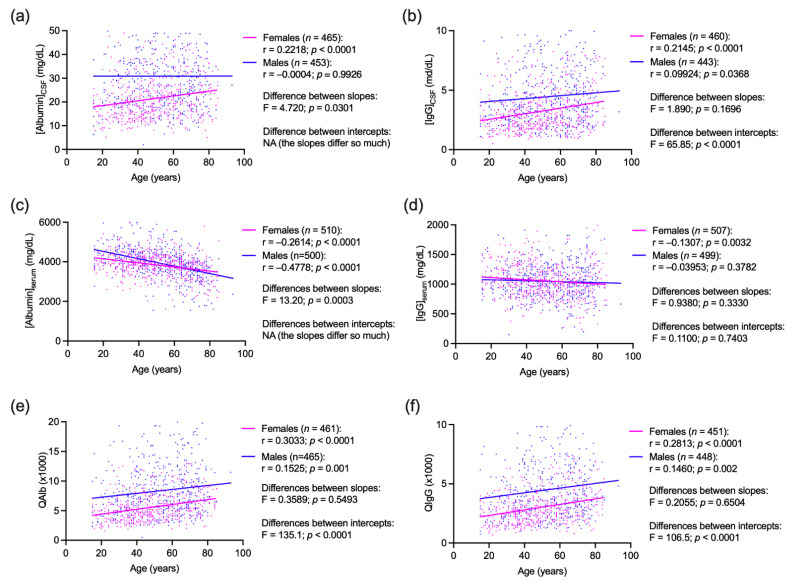
Linear regression analysis of age and (**a**,**b**) cerebrospinal fluid (CSF) and serum (**c**,**d**) concentrations of albumin and IgG and their (**e**,**f**) CSF/serum quotients (QAlb and QIgG respectively) in female and male patients. Presence of an intrathecal IgG synthesis was tested through the “gold standard” of oligoclonal bands (OCBs), and only subjects negative for the presence of OCBs were analyzed. Outliers were determined with robust regression and outlier removal (ROUT) method and excluded from any single analysis. Correlations were computed with Spearman test and F-test was used in regression analysis to compare the fits of linear models. NA: not applicable; when the slopes differed so much, it was not possible to test whether the intercepts differed significantly.

**Figure 2 diagnostics-11-00884-f002:**
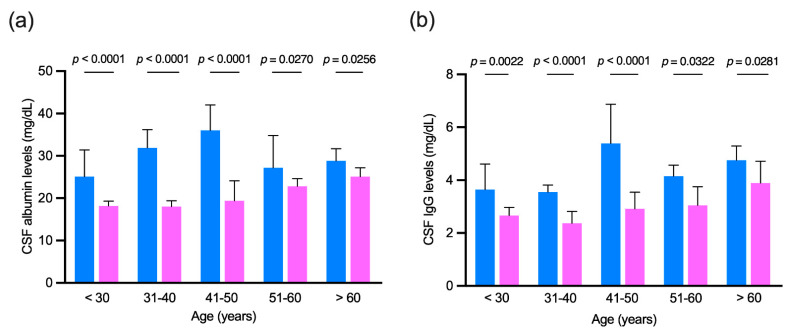
(**a**) Cerebrospinal fluid (CSF) albumin and (**b**) IgG levels in patients negative for the presence of intrathecal IgG synthesis grouped by age. Light blue, men; pink, women. Mann–Whitney test was used for all comparisons. Bars represent medians and 95% confidence intervals.

**Figure 3 diagnostics-11-00884-f003:**
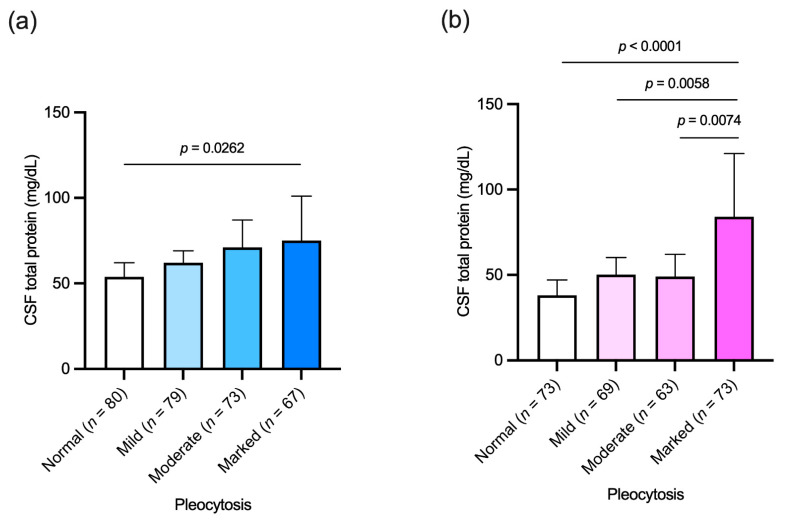
Cerebrospinal fluid (CSF) total protein levels in patients grouped by pleocytosis severity; normal (no more than 5 cells/μL); mild (6–25 cells/μL); moderate (26–100 cells/μL); marked (more than 100 cells/μL). (**a**) Men; (**b**) women. Kruskall–Wallis with Dunn’s multiple comparisons test were used in both analyses. Bars represent medians and 95% confidence intervals.

**Table 1 diagnostics-11-00884-t001:** Demographic characteristics and cerebrospinal fluid (CSF) reports of study population (cohort 1).

	Women (*n* = 859)	Men (*n* = 660)	*p*
Age, years: median (IQR)	43 (33–57)	50 (37–65)	<0.0001
CSF IgG (mg/dL): median (IQR)	3.59 (2.30–5.76)	4.63 (2.91–7.57)	<0.0001
CSF albumin (mg/dL): median (IQR)	19.2 (14.6–27.4)	29.7 (20.4–43.6)	<0.0001
Serum IgG (mg/dL): median (IQR)	1050 (876–1260)	1030 (861–1220)	0.1963
Serum albumin (mg/dL): median (IQR)	3940 (3630–4260)	4000 (3583–4408)	0.0866
Albumin quotient (QAlb) value: median (IQR)	4.93 (3.70–7.15)	7.50 (5.12–11.48)	<0.0001
IgG Index (positive): *n* (%)	309 (36.0)	160 (24.2)	<0.0001
Reiber’s formula (positive): *n* (%)	315 (36.7)	127 (19.2)	<0.0001
CSF IgG OCB patterns: *n* (%)			
Pattern 1 (negative):	445 (51.8)	429 (65.0)	<0.0001
Pattern 2 (local synthesis):	323 (37.6)	133 (20.2)	<0.0001
Pattern 3 (mixed):	24 (2.8)	23 (3.5)	0.4579
Pattern 4 (mirror):	57 (6.6)	61 (9.2)	0.0320
Pattern 5 (paraproteinemic):	10 (1.2)	14 (2.1)	0.1505

The comparison between categorical and continuous variables was made using the Fisher’s exact test and the Mann–Whitney test, respectively. Presence of IgG oligoclonal bands (OCBs) was assessed in all paired CSF and serum samples with the use of isoelectric focusing on agarose gel followed by IgG-specific immunoblotting. According to literature [9], we have identified the following CSF patterns: pattern 1 = absence of IgG OCBs; pattern 2 = 2 or more CSF-restricted IgG OCBs; pattern 3 = 2 or more CSF-restricted IgG OCBs with additional identical IgG OCBs in CSF and serum; pattern 4 = 2 or more identical IgG OCBs in CSF and serum; pattern 5 = some identical IgG OCBs in CSF and serum in a restricted pH range. Accordingly, only patterns 2 and 3 were considered suggestive of an intrathecal IgG synthesis. IQR: interquartile range.

**Table 2 diagnostics-11-00884-t002:** Agreement between the mathematical formulas and the “gold standard” of OCBs determination in evaluating the presence of an intrathecal IgG synthesis.

	Women (*n* = 859)		Men (*n* = 660)		
	Values	95% CI	Values	95% CI	*p*
**IgG Index**					
Kappa (SE)	0.587 (0.028)	0.531–0.642	0.559 (0.038)	0.485–0.633	
Sensitivity	0.7032	0.6531–0.7488	0.6731	0.5961–0.7418	0.5308
Specificity	0.8730	0.8414–0.8991	0.8909	0.8606–0.9152	0.3837
**Reiber’s Formula**					
Kappa (SE)	0.635 (0.027)	0.582–0.688	0.565 (0.0399)	0.488–0.642	
Sensitivity	0.7262	0.6770–0.7705	0.5962	0.5177–0.6699	0.0049
Specificity	0.8965	0.8671–0.9200	0.9325	0.9072–0.9513	0.0437

The agreement between the two methods has been evaluated using the Cohen’s Kappa test. Sensitivity and specificity in both sexes have been compared using the Fisher’s exact test. SE: standard error; CI: confidence interval.

**Table 3 diagnostics-11-00884-t003:** Cerebrospinal fluid (CSF) albumin and IgG levels were compared in women and in men negative for the presence of an intrathecal IgG synthesis with Quade statistics adjusting for age at lumbar puncture, and serum albumin or serum IgG levels.

	Women (*n* = 512)	Men (*n* = 504)	Mann–Whitney Statistics *p*	Quade Statistics *p*
CSFalbumin (mg/dL): median (IQR)	20.35 (14.90–30.08)	30.50 (20.13–45.05)	<0.0001	F = 83.329 <0.000001
CSF IgG (mg/dL): median (IQR)	2.93 (2.02–4.99)	4.21 (2.73–7.14)	<0.0001	F = 52.827 <0.000001

IQR: interquartile range.

**Table 4 diagnostics-11-00884-t004:** Spearman correlation analysis for cerebrospinal fluid (CSF) total protein levels and number of CSF white blood cells in a cohort of 424 neurological patients grouped by pleocytosis severity.

	Women	*p*	Men	*p*
**Pleocytosis:**				
Mild: 6–25 cells/μL (*n* = 148)	r = 0.1722 (*n* = 69)	0.1571	r = 0.1351 (*n* = 79)	0.2353
Moderate: 26–100 cells/μL (*n* = 136)	r = 0.02903 (*n* = 63)	0.8213	r = 0.1028 (*n* = 73)	0.3868
Marked: >100 cells/μL (*n* = 140)	r = 0.2013 (*n* = 73)	0.0876	r = 0.1172 (*n* = 67)	0.3448

## Data Availability

The datasets used and analyzed during the current study are available from the corresponding author on reasonable request.

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
