# Peer review of "Sex-Related Differences in Cerebrospinal Fluid Plasma-Derived Proteins of Neurological Patients"

_diagnostics, 2021, doi:10.3390/diagnostics11050884_

Round 1

Reviewer 1 Report

The major finding is that the male subjects have higher levels of serum-derived proteins, such as albumin and IgG in CSF without evidence to conclude its usefulness in diagnostic procedure. 

This retrospective analysis requires age-matched comparison of control and different disease groups or subgroups and further analyses of these groups with or without intrathecal IgG synthesis.

The major finding is that the male subjects have higher levels of serum-derived proteins, such as albumin and IgG in CSF without evidence to conclude its usefulness in diagnostic procedure. 

This retrospective analysis requires age-matched comparison of control and different disease groups or subgroups and further analyses of these groups with or without intrathecal IgG synthesis.

Reviewer 2 Report

This paper is an observational study on the sex related difference in CSF content in plasma related proteins such as albumin and IgG. It is s a well written paper focused on an important topic that is the diagnostic value of CSF in neurology, also on the positive side is the high number of samples that were analysed. However a major limitation is that it includes only pathological CSF from patients with neurological diseases so that it is difficult to draw conclusions on possible differences in physiological conditions.

Major amendments

  • the statement in the conclusions of abstract (page 1, line 28-29) is misleading "male subjects have  higher levels of serum derived proteins...."need to be corrected by specifying that the study is in pathological conditions. Also in the title should be mentioned that the results are related to pathological conditions
  • The differerences that are observed in CSF proteins may be related to different pathologies  that affect males versus females with different incidence rates. The information on the neurological diagnosis that were included and their relative frequence in this cohort of patients according to gender needs to be  reported and will greatly improve the quality of the results and discussion will be adequately integrated.
  • Plus in clinical settings the CSF routine analysis include the number of cells, integrating this data to the analysis would help in accounting for the role of inflammation/infection in altering the blood brain barrier.

 Minor points.

-page 2 line 48, B-CSF-B permeability, the acronym is not introduced plus it is steted that it is different but the underlying concept is omitted. Please integrate/clarify the sentence.

Round 2

Reviewer 1 Report

I do not have any further comment